# Prevalence of pulmonary tuberculosis among key and vulnerable populations in hotspot settings of Ethiopia. A systematic review and meta-analysis

**Melese Abate Reta**[1,2]*, **Zelalem Asmare**[1], **Assefa Sisay**[1], **Yalewayker Gashaw**[1], **Ermias Getachew**[1], **Muluken Gashaw**[1], **Zelalem Dejazmach**[1], **Abdu Jemal**[1], **Solomon Gedfie**[1], **Getinet Kumie**[1], **Marye Nigatie**[1], **Wagaw Abebe**[1], **Agenagnew Ashagre**[1], **Tadesse Misganaw**[1], **Woldeteklehaymanot Kassahun**[1], **Selamyhun Tadesse**[1], **Alene Geteneh**[1], **Atitegeb Abera Kidie**[3], **Biruk Beletew Abate**[4], **Nontuthuko Excellent Maningi**[5], **P. Bernard Fourie**[2]

1 Department of Medical Laboratory Science, College of Health Sciences, Woldia University, Woldia, Ethiopia, 2 Research Centre for Tuberculosis and Department of Medical Microbiology, Faculty of Health Sciences, University of Pretoria, Pretoria, South Africa, 3 Department of Public Health, College of Health Sciences, Woldia University, Woldia, Ethiopia, 4 Department of Nursing, College of Health Sciences, Woldia University, Woldia, Ethiopia, 5 Department of Microbiology, School of Life Sciences, College of Agriculture, Engineering and Science, University of Kwazulu Natal, Durban, South Africa

* melese1985@gmail.com

## Abstract

### Background

Despite the decline in tuberculosis (TB) incidence across many regions worldwide, including Ethiopia, the disease remains highly concentrated among vulnerable or socially marginalized populations and in high-risk settings. This systematic review and meta-analysis aims to estimate the pooled prevalence of pulmonary tuberculosis (PTB) among key and vulnerable populations (KVPs) residing in hotspot settings in Ethiopia.

### Methods

Potential papers were searched systematically in PubMed, Scopus, ScienceDirect databases, Google Scholar search engine, and institutional electronic repositories/registrars. A total of 34 potential articles that provide necessary information on the prevalence of PTB were reviewed and data were analyzed to determine the pooled prevalence of PTB among KVPs. The relevant data were recorded and analyzed using STATA 17.0. Cohen's kappa was computed to determine the agreement between reviewers, the Inverse of variance ($I^2$) to evaluate heterogeneity across studies, and Egger's test to identify publication bias. A random effect model was used to determine the pooled prevalence of PTB, subgroup analysis was computed by types of hotspot settings and year of publication.

**Data Availability Statement:** All relevant data are within the manuscript and its Supporting  information files.

**Funding:** The author(s) received no specific funding for this work.

**Competing interests:** The authors have declared that no competing interests exist.

**Abbreviations: MTB**, *Mycobacterium tuberculosis*; **PRISMA**, Preferred Reporting Items for Systematic Reviews and Meta-Analyses; **PROSPERO**, International Prospective Register of Systematic Reviews; **PTB**, Pulmonary tuberculosis; **TB**, Tuberculosis; **WHO**, World Health Organization.

## Results

This meta-analysis demonstrates that the pooled prevalence of PTB among populations residing in hotspot settings in Ethiopia was 11.7% (95% confidence interval (95CI): 7.97–15.43) with an $I^2$ of 99.91% and a $p < 0.001$. Furthermore, the subgroup analysis unveiled the pooled prevalence of PTB among KVPs residing in different hotspot settings as follows: Prison inmates 8.8% (95CI: 5.00–12.55%), University students 23.1% (95CI: 15.81–30.37%), Refugees 28.4% (95CI: -1.27–58.15%), Homeless peoples 5.8% (95CI: -0.67–12.35%), Healthcare settings 11.1% (95CI: 0.58–21.63%), Spiritual holy water sites attendees 12.3% (95CI: -6.26–30.80%), and other high-risk settings 4.3% (95CI: 0.47–8.09%). Besides, the subgroup analysis revealed that the pooled prevalence of PTB post-2015 was 10.79% (95CI: 5.94–15.64%), whereas it stood at 14.04% (95CI: 10.27–17.82%) before 2015.

## Conclusion

The prevalence of PTB among KVPs residing in the hotspot settings in Ethiopia remains significant, with a weighted pooled prevalence of 11.7%. Thus, the national TB control programs should give due attention and appropriate control measures should be instituted that include regular systematic TB screening, compulsory TB testing for presumptive TB cases among KVPs, and tightened infection control at hotspot settings.

## Introduction

Tuberculosis (TB), caused by the human-adapted *Mycobacterium tuberculosis* (Mtb), remains an important public health concern worldwide, leading to considerable morbidity and mortality [1, 2]. TB is responsible for the loss of over 1.5 million lives and infects about 10.0 million people each year [1, 2]. Despite the declining tendency in TB cases globally, including in Ethiopia, the disease persists predominantly among vulnerable or socially marginalized populations and in those who reside in a high-risk environment [1, 3, 4].

While Ethiopia has transitioned out of the list of 30 countries with the highest burden of multidrug-resistant (MDR) TB, it continues to be classified as one of the thirty countries with high TB incidence [1]. Ethiopia's TB incidence was 126 per 100,000 population, in 2022 [2]. Ethiopia aims to meet the global target of identifying 90% of TB cases among high-risk populations by 2025 [5]. Thus, in 2017, the Ethiopian National TB Control Program developed an operational guide and implementation strategy especially targeting key populations affected by TB [4], despite the slowing pace of implementation and efforts in the country.

KVPs for TB are identified by their higher risk due to biological and socioeconomic factors, limited healthcare access, and human rights violations [3, 4, 6]. Despite the fact that country-specific conditions are important to define KVPs for TB [3, 7], Ethiopia has identified KVPs, including people living with human immunodeficiency virus (HIV), those with diabetes, children, older adults, prison inmates, University residents, contacts of TB patients, refugees, internally displaced people, homeless individuals, female sex workers, and healthcare workers [4]. While Ethiopia's TB prevention and control program integrates and enforces infection control and transmission prevention measures in healthcare facilities, congregate settings, and households, the incidence of TB remains disproportionately high among KVPs residing in

hotspot areas within the country [8]. Ensuring access to KVPs in the TB response is equitable, upholds human rights, and is crucial for countries aiming to end the TB epidemic [7]. Several studies were conducted throughout the country to determine the prevalence o PTB among residents of various high-risk environments [9–21]. Thus, in Ethiopia, significantly higher prevalence of TB was reported among populations residing in prisons (8.3–9.84%) [9, 10], refugee campus (13.3–43.5%) [11, 12], University residences (14.0–30.13%) [13–15], spiritual holy water sites (2.9–21.8%) [16, 17], healthcare settings (1.6–13.7%) [18, 19], and homeless shelters (2.6–9.2%) [20, 21]. Furthermore, an earlier large-scale community-based study aimed at investigating the prevalence of TB among KVPs in Ethiopia reported a prevalence rate of 4.3% [3]. Additionally, a previous study conducted to examine the prevalence of PTB among attendees of spiritual holy water sites also found that the prevalence of bacteriologically confirmed PTB was 4.3 times higher [22] than in a national TB prevalence report in Ethiopia [23]. Hence, to enhance TB prevention and control measures, it is imperative to identify KVPs within the community, undertake systematic TB screening, and comprehend TB prevalence rates [24]. These efforts support the national TB prevention and control program in giving due attention, prioritizing actions, and designing evidence-based interventions [24]. While the transmission rate and conditions for TB transmission differ among populations residing in high-risk or hotspot settings, there is limited evidence regarding the pooled prevalence of PTB among KVPs in Ethiopia. To the best of our knowledge, only two systematic reviews and meta-analyses have been undertaken to estimate the weighted pooled prevalence of PTB among prison inmates in Ethiopia [9, 10]. Therefore, this systematic review and meta-analysis aims to determine the weighted pooled prevalence of pulmonary tuberculosis among KVPs residing in the hotspot settings in Ethiopia.

## Methods

### The study protocol and registration

The protocol for this systematic review and meta-analysis was thoroughly developed and registered on the International Prospective Register of Systematic Reviews (PROSPERO) (CRD42024543925).

### Databases and article search strategies

A systematic article search was conducted in PubMed, Scopus, ScienceDirect electronic databases, Google Scholar search engines, and institutional electronic repositories/registrars. Key search words/terms, like Medical Subject Headings (MeSH) terms and other keywords utilized to retrieve potential English papers published within the timeframe of 01 January 2000 to 31 December 2023. The main search terms/words or phrases were: "Prevalence" OR "epidemiology" AND "tuberculosis" OR "TB" OR "pulmonary tuberculosis" OR "*Mycobacterium tuberculosis*" OR "*M. tuberculosis*" OR "MTB" AND "congregate settings" OR "high-risk settings" OR "hotspot settings" OR "Spiritual holy water sites" OR "prison" OR "prisoner" OR "prison inmate" OR "jail" OR "correctional facilities" OR "Universities" OR "Homeless shelters" OR "Homeless" OR "Military camps" OR "Refugee" OR "Refugee camps" OR "Internally displace people" OR "Nursing homes" OR "healthcare workers" OR "healthcare settings" AND Ethiopia. The full search strategy and search terms for the PubMed/MEDLINE database are depicted in the S1 Table in S1 File. Moreover, we reviewed the reference lists of primary studies and review papers to identify grey literature that was not catching up through our systematic review.

## Eligibility criteria

All studies retrieved from the searched databases and other sources were imported to a reference manager software, EndNote X7 (Thomson Reuters, Toronto, Ontario, Canada). During this systematic review, we followed the updated Preferred Reporting Items for Systematic Reviews and Meta-Analyses (PRISMA) [25]. Duplicate papers were removed using automation, EndNote software, and manually, and the retained studies were subjected to initial screening by title and abstract, followed by detailed full-text screening by three reviewers (MAR, ZA, and AS) independently. To identify and retain potential/eligible articles, we formulate clear predetermined inclusion and exclusion criteria. Articles that fulfill the inclusion criteria were reviewed and included, whereas papers that did not fulfill the inclusion criteria were excluded: **Inclusion criteria**: (a) *Country and setting*: Ethiopian hotspot settings (i.e., prisons, refugee camps, Universities, healthcare settings, spiritual holy water sits, homeless shelters, etc); (**b**) *Study subjects*: prison inmates, refugees, University students and residents, healthcare workers, holy water site attendees, homeless people, internally displaced peoples; (**c**) *Study design*: A cohort study (prospective and retrospective), cross-sectional, case-control, and randomized control trial; (**d**) Reported the sample size, PTB cases, or prevalence; (**e**) Peer-reviewed, and published in English; (**f**) Undertook experimental or retrospective works; (g) Reported types of laboratory tests and specimens used for diagnostic; (**h**) Reported type of TB identified and types of TB patients; (**i**) Reported quality control/assurance measures. **Exclusion criteria: (a)** Studies conducted outside Ethiopia; (**b**) Reports TB prevalence in other patients; (**c**) Studies reported only extrapulmonary TB cases or latent TB; (**d**) Studies reported only TB patients' treatment outcomes; (**e**) Groups or not linked to TB prevalence in prison, refugees, Universities, healthcare settings, spiritual holy water settings, homeless shelters of Ethiopia; (**f**) Reported the knowledge, attitude, and practice of prisoners, refugees, Universities students, healthcare workers, spiritual holy water site attendees, homeless individuals toward TB disease; (**g**) Studies published in other languages; (**h**) Investigated drug resistance patterns only; (**i**) Incomplete data and inaccessible full-text articles after three emails communications from the corresponding author; (**j**) low-quality articles, duplicate publications, or extensions of analysis from original studies were excluded.

## Quality assessments

All potential papers were assessed for quality using the recommended quality assessment tool, the Joanna Briggs Institute (JBI), for prevalence data/observational studies [26]. Two reviewers (MAR and ZA) independently performed a critical appraisal of each paper. This involved a thorough examination using the JBI appraisal domains as follows: (**a**) appropriate sample frame to address target population; (**b**) appropriate sampling of study subjects; (**c**) adequate sample size; (**d**) detailed description of study subjects and settings; (**e**) data analysis with sufficient coverage of samples and valid method used for the identification of the condition; (**f**) standard and reliable measurement for the condition; (**g**) appropriate statistical analysis; and (**h**) adequate response rate, or appropriate management of the low response rate. If consensus between the two independent reviewers could not be reached, a third reviewer (AS) was engaged to resolve any disagreements and facilitate consensus. Studies scoring 50% or higher on the final quality assessment were eligible for inclusion in this systematic review and meta-analysis (S2 Table in S1 File).

## Data extraction from included studies

A standardized data extraction format was employed using a Microsoft Excel sheet to document pertinent information from each potential study included. The extraction process

covered various domains, including study characteristics, such as the first author's surname, study period, year of publication, study geographical location, study setting, study design, TB diagnostics method(s), inclusion criteria, specimen type, sample size, types of TB identified, number of TB cases, and prevalence of PTB (Table 1).

## Study outcomes

The main outcomes of interest to our systematic review and meta-analysis were the pooled prevalence of PTB among the KVPs residing in the hotspot settings or risk settings for TB transmission in Ethiopia, as well as the subgroup-weighted pooled prevalence estimate of PTB in populations residing at each hotspot setting (i.e. prison, refugee camps, healthcare settings, universities, homeless shelters, and holy water sites, and other high-risk settings). We also computed the trends of PTB to examine the prevalence before and post-2015. If the eligible and included studies report both the incidence or prevalence of PTB and extrapulmonary TB (EPTB) among presumptive TB patients, we carefully extract the data pertaining to PTB cases only and calculate its prevalence.

## Operational definition

*Hotspot settings/high-risk settings*: It defined as places where populations gather for extended periods, creating favorable conditions for the transmission of PTB (i.e., prisons, refugee camps, homeless shelters, University residences, healthcare settings, spiritual holy water sites, etc) [4, 24]. **Key and vulnerable populations**: Refers to those residing in hotspots or high-risk settings for TB transmission. This includes prison inmates, refugees, homeless individuals, University students, healthcare workers, attendees of spiritual holy water sites, internally displaced people, and others with a high vulnerability to TB infection and developing the disease, such as HIV patients, individuals with diabetes, children, and older adults [3, 4, 24].

## Data management and statistical analysis

The recorded data from the included potential studies using a pre-prepared Microsoft Excel sheet was double-checked for its accuracy, and then it was exported to STATA 17.0 software (StataCorp, Texas, USA) for the final analysis. The heterogeneity across studies was assessed using the inverse variance ($I^2$) test, with interpretations assigned to $I^2$ values as follows: 0.0% (no heterogeneity), 0–25% (low heterogeneity), 25–50% (medium heterogeneity), and >75% (high heterogeneity) [27]. Furthermore, we assessed the source of heterogeneity across studies using Galbraith's visual inspection. A subgroup analysis, considering various categories, was conducted for studies showing high heterogeneity. The presence of publication bias was assessed using Egger's test, with a significance threshold set at $p < 0.05$. Additionally, a trim-and-fill analysis was carried out to identify and address any potential publication bias. A random effect model for meta-analysis was used to estimate the pooled prevalence of PTB among KVPs in hotspot settings in Ethiopia. When calculating the pooled prevalence of PTB among KVPs in hotspot settings, if studies reported a prevalence of zero or one hundred percent, resulting in a zero standard error, we applied continuity correction as recommended [28].

# Results

## Searching results

In this systematic review and meta-analysis, a total of 8,546 studies were retrieved from searched electronic databases, and other sources, such as Google Scholar and registrars. Of the total, 4,247 articles were non-duplicated and subjected to further evaluation; and 4,185 were

**Table 1. Characteristics of included studies.**

| Author's Name | Year of Publication | Region | Study design | Study setting(s) | TB diagnostics method/s | Inclusion criteria | Sample size | Specimen | Types of TB identified | # of cases | Prevalence (%) | Point prevalence per 100,000 pop | Quality score |
|---|---|---|---|---|---|---|---|---|---|---|---|---|---|
| Moges et al [29] | 2012 | AM | CS | Prison | LED microscopy, Cytology | Cough ≥1 week | 250 | Sputum, FNAC | SPPTB | 26 | 10.40 | 1482.3 | 8 |
| Abebe et al [30] | 2011 | MIXED (Dire-Dawa, Somali, Harari) | CS | Prison | ZN Microscopy & Culture | Cough ≥2 weeks | 382 | Sputum* | PTB | 44 | 11.52 | 1913 | 8 |
| Addis et al [31] | 2015 | AM | CS | Prison | ZN Microscopy | Cough ≥2 weeks | 384 | Sputum* | SPPTB | 33 | 8.59 | 2032 | 7 |
| Bayu et al [32] | 2016 | SNNP | CS | Prison | ZN Microscopy | Cough ≥2 weeks | 305 | Sputum* | PTB | 17 | 5.57 | 966 | 6 |
| Fuge et al [33] | 2016 | SNNP | CS | Prison | ZN Microscopy | Cough ≥1 week | 164 | Sputum | SPPTB | 3 | 1.83 | 349.2 | 5 |
| Zerihun et al [34] | 2014 | SNNP | CS | Prison | ZN Microscopy & Culture | Cough ≥2 weeks | 124 | Sputum | PTB | 24 | 19.35 | 629 | 6 |
| Biadglegne et al [35] | 2014 | AM | CS | Prison | ZN Microscopy, Culture, Xpert MTB/RIF | Cough ≥1 week, able to produce, prison inmates | 207 | Sputum* | SNPTB | 23 | 11.11 | | 8 |
| Ali et al [36] | 2015 | MIXED (Oromia, Gurage, SNNP, Harar) | CS | Prison | ZN Microscopy & Culture | WHO grade 5 TB screening criteria**, ≥18yrs & either HIV+, treatment in the last 5yrs, | 765 | Sputum | PTB | 71 | 9.28 | 458.2 | 9 |
| Gebrecherkos et al [37] | 2016 | AM | CS | Prison | ZN & LED Microscopy, Xpert MTB/RIF | Cough ≥2 weeks, not-on anti-TB treatment | 282 | Sputum | SPPTB | 15 | 5.32 | 384.6 | 9 |
| Adane et al [38] | 2016 | TG | CS | Prison | ZN Microscopy & Culture | WHO grade 5 TB screening criteria**, ≥18yrs, not-on anti-TB treatment | 1258 | Sputum* | PTB | 74 | 5.88 | 793.5 | 8 |
| Winsa et al [39] | 2015 | OR | CS | Prison | ZN Microscopy | Cough ≥2 weeks | 196 | Sputum | PTB | 43 | 21.94 | | 5 |
| Gizachew et al [40] | 2017 | AM | CS | Prison | Xpert MTB/RIF | Cough ≥2 weeks | 265 | Sputum | PTB | 9 | 3.40 | | 7 |
| Merid et al [41] | 2018 | SNNP | CS | Prison | ZN Microscopy & Xpert MTB/RIF | Cough ≥2 weeks | 372 | Sputum | PTB | 31 | 8.33 | 1748 | 8 |
| Berihun et al [42] | 2018 | AM | RCS | Prison | AFB Smear Microscopy | Prison inmates, on-anti-TB treatment | 162 | Sputum* | PTB | 73 | 45.06 | 2139 | 6 |

(*Continued*)

**Table 1.** (Continued)

| Author's Name | Year of Publication | Region | Study design | Study setting (s) | TB diagnostics method/s | Inclusion criteria | Sample size | Specimen | Types of TB identified | # of cases | Prevalence (%) | Point prevalence per 100,000 pop | Quality score |
|---|---|---|---|---|---|---|---|---|---|---|---|---|---|
| Abayineh [43] | 2018 | AA | CS | Prison | ZN Microscopy & Xpert MTB/RIF | Cough ≥2 weeks, prison inmates, >18yrs of age | 218 | Sputum | PTB | 11 | 5.05 | | 7 |
| Agajie et al [44] | 2018 | BGR | CS | Prison | Xpert MTB/RIF | Cough ≥2 weeks, ≥ 18yrs of age | 3395 | Sputum | PTB | 8 | 0.24 | 236 | 7 |
| Tsegaye et al [45] | 2019 | AA | CS | Prison | ZN Microscopy, CXR, Xpert MTB/RIF & Culture | prison inmates (> 18yrs of age) | 13803 | Sputum | BC-PTB | 22 | 0.16 | | 8 |
| Dibissa et al [46] | 2019 | OR | CS | Prison | ZN Microscopy, Culture & Xpert MTB/RIF | Prison inmates, cough ≥2 weeks | 270 | Sputum | PTB | 42 | 15.56 | 744 | 7 |
| Adane et al [47] | 2019 | MIXED (Amhara, Tigray) | CRT | Prison | ZN Microscopy, CXR, Xpert MTB/RIF & Culture | Patients up to 15 years old; have presumptive DR-TB EPTB, or their HIV(+) or unknown | 1124 | Sputum | MTB | 34 | 3.02 | | 8 |
| Duressa et al [48] | 2022 | BGR | CS | Prison | Xpert MTB/RIF | Prison inmates with cough ≥2 weeks; or known HIV (+) with able/unable to produce sputum; WHO grade 5 TB screening criteria** | 212 | Sputum | PTB | 5 | 2.36 | 279 | 7 |
| Dememew et al [3] | 2020 | MIXED (Harar, Dire Dawa, Amhara, Oromia) | CS | Key people at hotspot settings (FSWs, IDPs, IMWs, RMC, HCWs) | AFB smear Microscopy, Xpert MTB/RIF, FNA, CXR/ clinical evaluation | WHO grade 5 TB screening criteria** | 1,878 | Sputum | SPPTB + SNPTB | 87 | 4.63 | 1,519 | 8 |
| Derseh et al [16] | 2017 | AM | CS | Holy water sites | Auramine O staining, Fluorescent Microscopy | Attendees ≥ 15yrs of age, WHO grade 5 TB screening criteria** | 382 | Sputum | PTB | 11 | 2.88 | 795 | 9 |
| Hordofa et al [49] | 2023 | OR | CS | Prison | Xpert MTB/RIF | Prisoners with cough≥2 weeks, chronic patients or a history of TB treatment > 1 week | 259 | Sputum | PTB | 14 | 5.41 | | 9 |

*(Continued)*

Table 1. (Continued)

| Author's Name | Year of Publication | Region | Study design | Study setting (s) | TB diagnostics method/s | Inclusion criteria | Sample size | Specimen | Types of TB identified | # of cases | Prevalence (%) | Point prevalence per 100,000 pop | Quality score |
|---|---|---|---|---|---|---|---|---|---|---|---|---|---|
| Meaza et al [12] | 2023 | MIXED (Tigray, Afar, Gambella, Benishangul-gumuz, Somali) | CS | Refugee camps | Xpert MTB/RIF, ZN Microscopy, LJ & MGIT960 culture | All refugees who had TB-like symptoms or contact history with active TB patients, and were referred to refugees clinical for diagnosis | 610 | Sputum | BC-PTB | 81 | 13.28 |  | 9 |
| Mekonen et al [13] | 2018 | MIXED (Oromia, Dire-Dawa, Somali) | CS | University | ZN Microscopy & LJ culture | All students with cough > 2 weeks, WHO grade 5 TB screening criteria** | 1097 | Sputum | SPPTB | 153 | 13.95 | 433 | 8 |
| Mekonen et al [14] | 2016 | MIXED (Addis Ababa & Oromia) | RCS | University | ZN Microscopy | - | 375 | Sputum | SPPTB | 113 | 30.13 |  | 7 |
| Moges et al [15] | 2015 | AM | RCS | University | ZN Microscopy, Culture & radiology | All TB cases diagnosed with smear, culture, and/or radiography were included in the study. | 181 | Sputum | SPPTB | 46 | 25.41 |  | 7 |
| Semunigus et al [20] | 2016 | AM | CS | Homeless | Fluorescence Microscopy & Xpert MTB/RIF | Homeless individuals ≥ 15yrs of age, cough ≥ 2 weeks | 351 | Sputum | SPPTB | 9 | 2.56 | 505 | 8 |
| Shamebo et al [21] | 2023 | AA | CS | Homeless | Xpert MTB/RIF & LJ culture | WHO TB symptoms screening criteria** | 641 | Sputum | PTB | 59 | 9.20 | 1053.6 | 8 |
| Shiferaw et al [18] | 2021 | AM | CS | Healthcare settings | Xpert MTB/RIF | Participants with cough ≥ 2 weeks, WHO TB symptoms screening criteria**, and Healthcare workers who worked in the Amhara region. Both clinical, administrative & support staff. | 580 | Sputum | PTB | 9 | 1.55 | 1600 | 9 |
| Wolde et al [50] | 2017 | OR | CS | University | ZN Microscopy & MGIT 960 culture | Cough ≥2 weeks | 129 | Sputum | PTB | 31 | 24.03 | 209.2 | 6 |
| Legesse et al [11] | 2021 | MIXED (Gambella, Afar, Benishangul, SNNPs, Tigray) | RCS | Refugee camps | ZN Microscopy | WHO TB symptoms screening criteria** | 1553 | Sputum | PTB | 677 | 43.59 |  | 8 |

(Continued)

**Table 1.** (Continued)

| Author's Name | Year of Publication | Region | Study design | Study setting (s) | TB diagnostics method/s | Inclusion criteria | Sample size | Specimen | Types of TB identified | # of cases | Prevalence (%) | Point prevalence per 100,000 pop | Quality score |
|---|---|---|---|---|---|---|---|---|---|---|---|---|---|
| Reta et al [17] | 2023 | AM | CS | Holy water sites | LJ culture | Attendees ≥ 18yrs with cough ≥ 2weeks, WHO and National TB symptoms screening criteria** | 560 | Sputum | PTB | 122 | 21.79 | 1183 | 8 |
| Eyob et al [19] | 2002 | AA | RCS | Healthcare settings | BC-PTB | All workers employed at the TB demonstration and training center | 175 | | PTB | 24 | 13.71 | 13714.3 | 5 |

*The number of prisoners who were on anti-TB treatment were added to the sample size as well as to the reported numbers of cases to estimate prevalence;

** = WHO grade 5 TB identification criteria: cough ≥2 weeks, sputum production, chest pain, loss of appetite, weight loss in last 3 months.

**Abbreviations**: **AA**: Addis Ababa; **AM**: Amhara; **BC-PTB**: bacteriologically confirmed pulmonary tuberculosis; **BGR**: Benishangul-Gumuz; **CS**: Cross-sectional; **CXR**, chest x-ray; **DR-TB**: Drug-resistance; **EPTB**: Extra-pulmonary tuberculosis; **FNAC**: Fine needle aspiration cytology; **FSWs**: Female sex workers; **HCWs**: Healthcare Workers; **HIV**: Human immunodeficiency virus; **IDPs**: Internally displaced peoples; **IMWs**: Internal migratory workers; **LED**: Light emitting diode; **LJ**: Lowenstein Jenssen; **MGIT**: Mycobacterium growth indicator tube; **OR**: Oromia; **PTB**: pulmonary tuberculosis; **RCS**: Retrospective cross-sectional; **RCT**: Randomize clinical trial; **RMC**: Residents of missionary charity; **SNNP**: Southern nations, nationalities and peoples; **SNPTB**: Smear negative pulmonary tuberculosis; **SPPTB**: Smear positive pulmonary tuberculosis; **TB**: Tuberculosis; **TG**: **Tigray**; **WHO**: World Health Organization; **ZN**: Ziehl-Neelsen.

assessed and excluded after reviewing their title, abstract, and for other reasons (duplicate studies, review, studies only report TB treatment outcome, non-English papers, and others), while 62 papers were retained for full-text evaluation. After a full-text review, the final meta-analysis included 34 potential articles [3, 11–21, 29–50] that reported on the prevalence of PTB among KVPs residing in the hotspot settings in Ethiopia (Fig 1).

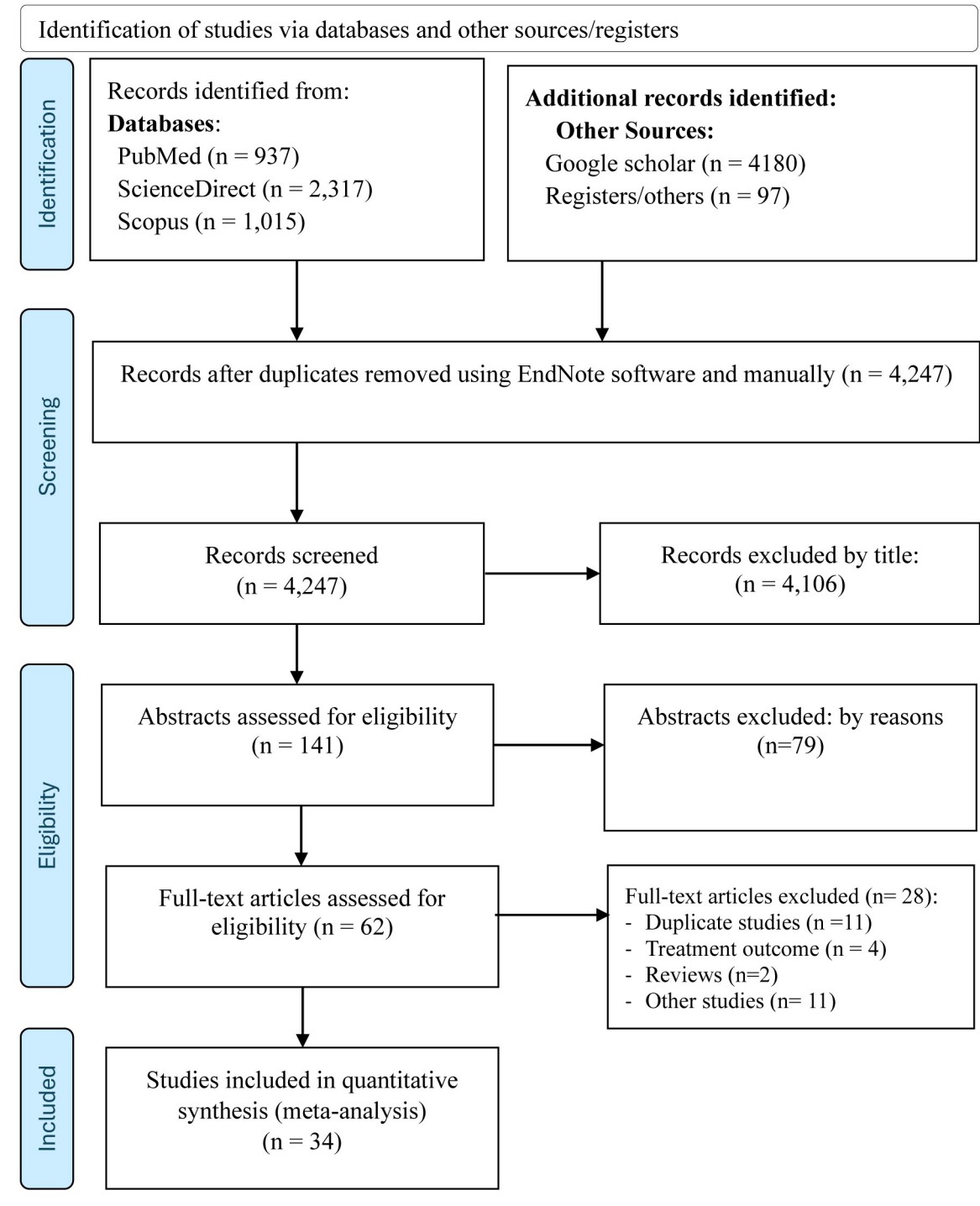

**Fig 1. PRISMA flow diagram shows the results of the search and reasons for exclusion [25].**

## Descriptive summary of included studies

This systematic review and meta-analysis retrieved and included a total of 34 potential papers that fulfill the inclusion criteria and provided necessary data on the prevalence of PTB among KVPs residing in the hotspot settings in Ethiopia. In this review, a total of 32,909 participants with presumptive TB, resulting in 2,044 (6.2%) PTB cases were involved. Of the included studies, 21 studies were conducted in prison settings [29–49], 4 studies in University students [13–15, 50], while the remaining (n = 2) [11, 12] in refugee camps, homeless shelters (n = 2) [20, 21], spiritual holy water sites (n = 2) [16, 17], and healthcare settings (n = 2) [18, 19]. Almost all studies were cross-sectional by study design [3, 12, 13, 16–18, 20, 21, 29–41, 43–46, 48–50], except (n = 5) were retrospective cross-sectional [11, 14, 15, 19, 42], and (n = 1) [47] was clustered randomized trials. Furthermore, twenty-five studies were published after 2015 [3, 11–14, 16–18, 20, 21, 32, 33, 37, 38, 40–50], while nine studies were published before 2015 [15, 19, 29–31, 34–36, 39]. For diagnosis of the patients and bacteriological confirmations, smear microscopy alone (using either light-emitting diode fluorescent microscope or Ziehl-Neelsen), or in combination with the GeneXpert®MTB/RIF assay and rarely culture techniques were utilized. During the screening of the patient/participants, almost all the studies followed the World Health Organization (WHO) and national TB symptoms screening guidelines [51, 52]. The prevalence of PTB among these high-risk populations residing in the hotspot setting ranged from 0.16% to 45.1% [42, 45]. The characteristics of the studies included are summarized in Table 1.

## Meta-analysis results

**Pooled prevalence of pulmonary TB among KVPs.** This systematic review retrieved 34 potential papers with necessary information on PTB prevalence among KVPs in Ethiopia's hotspot settings [3, 11–21, 29–50]. This meta-analysis resulted in a pooled prevalence of PTB at 11.7% (95%CI: 7.97–15.43) with an $I^2$ of 99.91% and a *p*-value of < 0.001(Fig 2). During our assessment of the existence of publication bias, the funnel plot showed asymmetrical distribution (S1 Fig in S1 File), indicating the presence of publication bias. Also, for Egger's regression test, the *p*-value was <0.00001(S3 Table in S1 File), indicating the presence of publication bias across studies. To correct the publication bias, the meta-trim-and-fill analysis was performed, resulting in no change in the pooled prevalence of PTB (S4 Table in S1 File).

**Subgroup analysis by types of hotspot settings.** During this meta-analysis to estimate the weighted pooled prevalence of PTB among the KVPs residing in the hotspot settings, heterogeneity across studies was observed. Due to the existence of high heterogenicity ($I^2$ = 99.91%; *p*<0.001) (Fig 2 and S2 Fig in S1 File), we conducted the subgroup analysis by types of hotspot settings and year of publication. Fig 3 depicts the weighted pooled prevalence of PTB among KVPs residing in different high-risk/hotspot settings. This meta-analysis revealed that the pooled prevalence of PTB in different hotspot settings was as follows: Prison inmates 8.8% (95%CI: 5.00–12.55), University students 23.1% (95%CI: 15.81–30.37), Refugee camps 28.4% (95%CI: -1.27–58.15), Homeless individuals 5.8% (95%CI: -0.67–12.35), Healthcare settings 11.1% (95%CI: 0.58–21.63), Spiritual holy water sites 12.3% (95%CI: -6.26–30.80), and other hotspot settings 4.3% (95%CI: 0.47–8.09). Although sufficient studies were not found to generate robust evidence, the pooled prevalence of PTB among University students (23.1%) and refugees (28.4%) was notably higher (Fig 3).

**Subgroup analysis by year of publication.** This meta-analysis demonstrated that the pooled prevalence of PTB among these KVPs residing in a high-risk setting steadily decreased after 2015, despite the fact that the majority of the included studies were

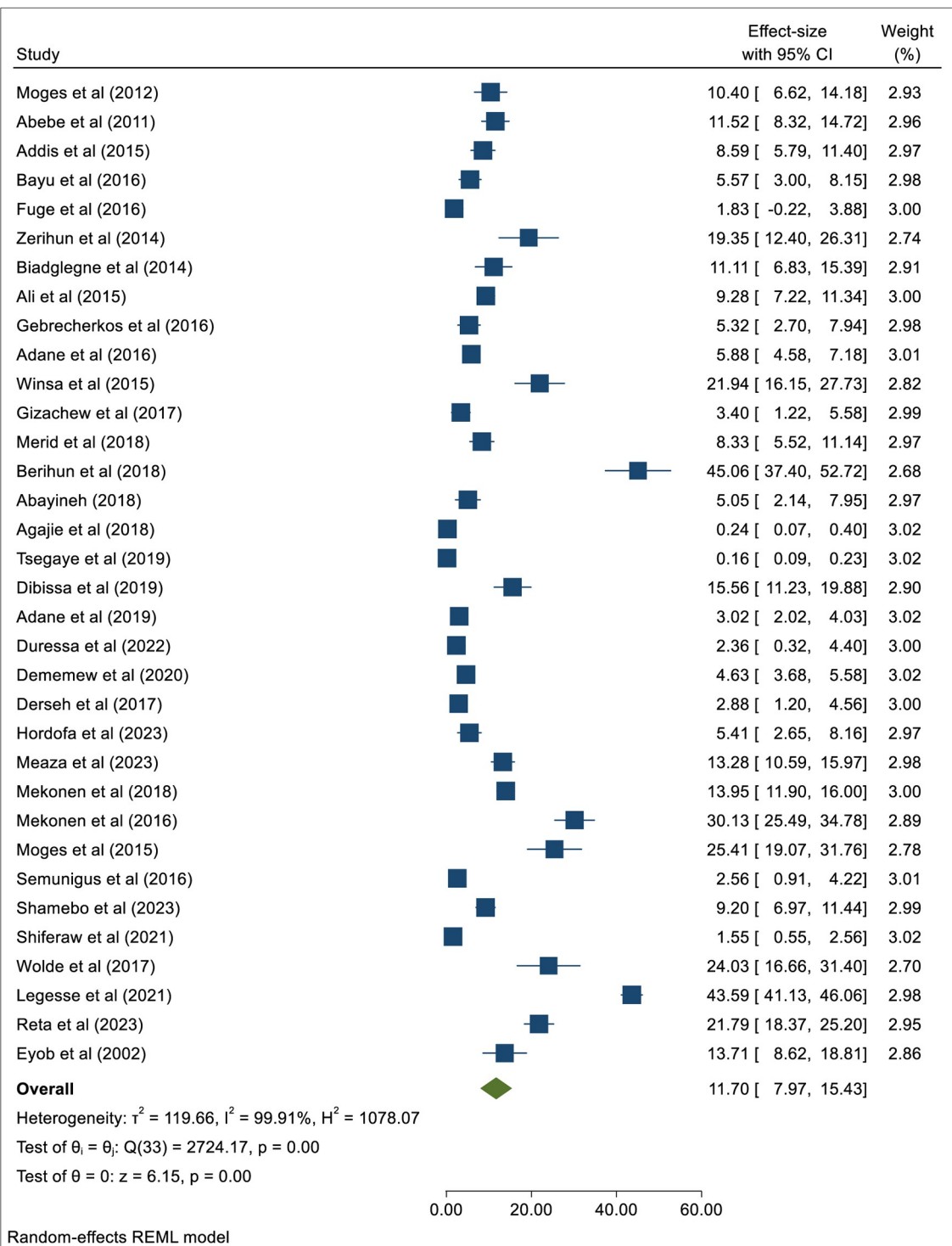

**Fig 2. Forest plot showing the pooled prevalence of PTB among the KVPs residing in the hotspot settings in Ethiopia.**

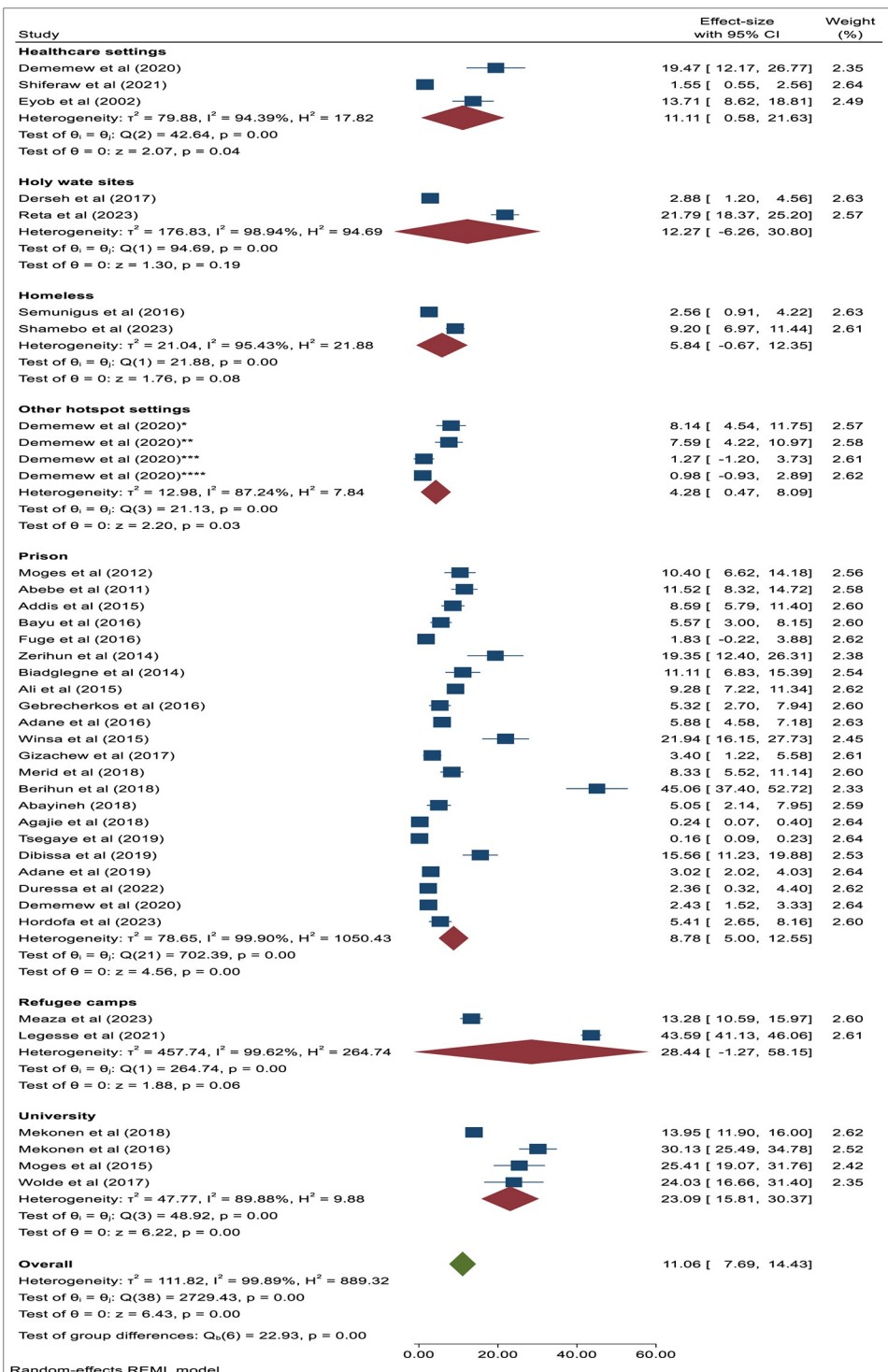

**Fig 3. The forest plot showing the subgroup analysis of the pooled prevalence of PTB by types of hotspot settings in Ethiopia. Other hotspot settings:** * = Female sex workers, ** = Internal migratory workers, *** = Residents of missionary charity, **** = Internally displaced peoples.

published after 2015. Thus, the combined prevalence of PTB among KVPs post-2015 was 10.79% (95% CI: 5.94–15.64) with an $I^2$ of 99.95% and a *p*-value of < 0.001, whereas it stood at 14.04% (95% CI: 10.27–17.82) with an $I^2$ of 88.38% and a *p*-value of < 0.001 before 2015 (Fig 4).

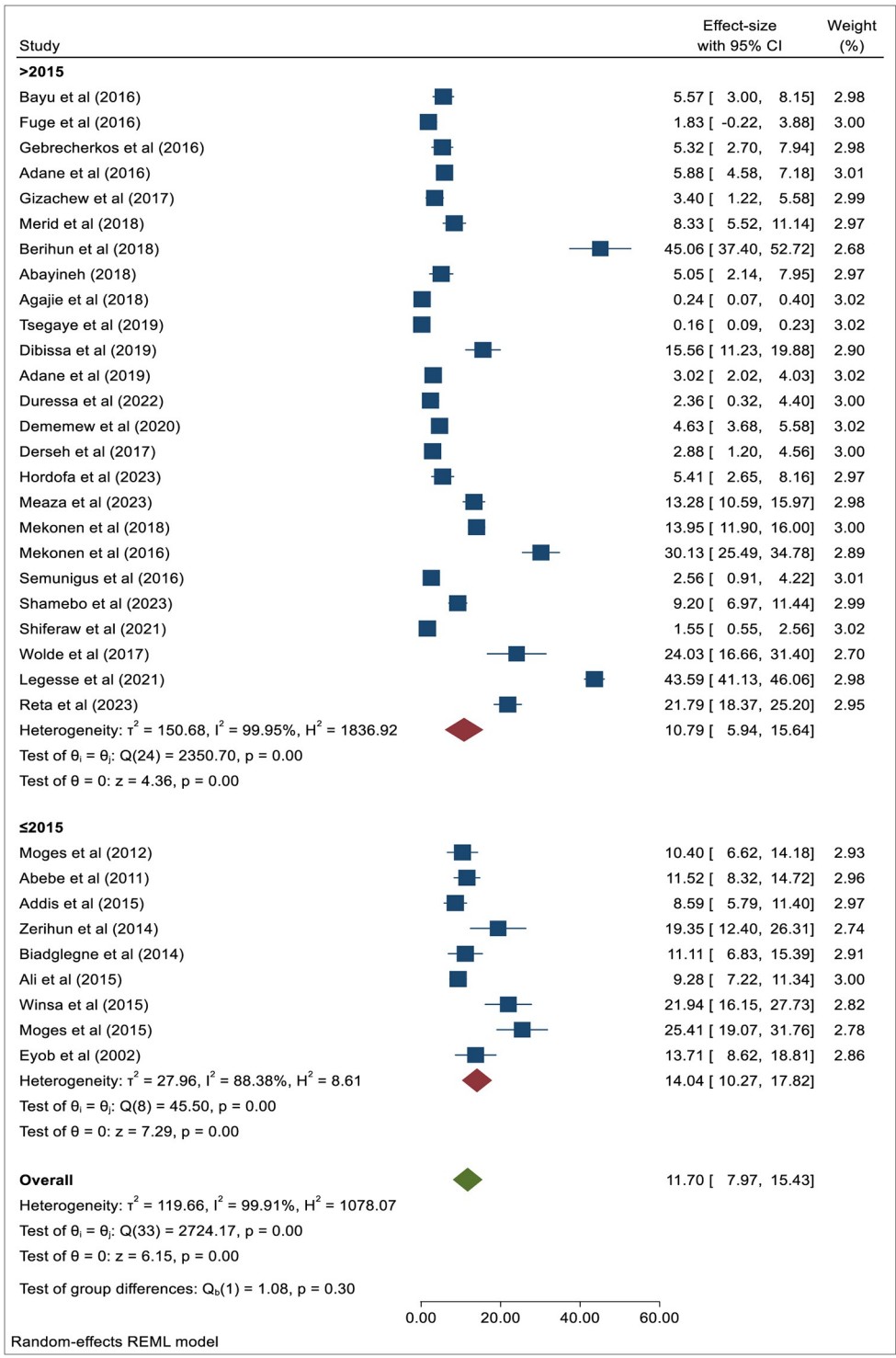

**Fig 4. The forest plot showing the subgroup analysis of the pooled prevalence of PTB by year of publication.**

## Discussion

Although TB incidence is declining in many regions globally, the disease remains prevalent, particularly among vulnerable or socially marginalized populations and those residing in high-risk environments [1, 3, 4, 6]. *Mycobacterium tuberculosis* is an airborne infection, so it is commonly believed that transmission primarily happens indoors [53]. Thus, due to the over-crowded living conditions, poor ventilation, and compromised health status of individuals living in high-risk or hotspot settings such as prisons, refugee camps, holy water sites, and homeless shelters, the transmission and prevalence of pulmonary TB are higher than in the general population [3, 6, 9, 10, 16, 17, 54]. The current systematic review and meta-analysis was designed to provide essential epidemiological data that is important for TB control initiatives within the nation and to support global efforts to control the disease. Furthermore, it gives crucial information to guide specific and cost-effective TB control, which is a topic of discussion globally. This study provides valuable information regarding TB, encompassing its overall prevalence at the national level, with a specific emphasis on KVPs in hotspot settings, and its magnitude across various hotspot settings and timeframes.

In this meta-analysis, the weighted pooled prevalence of pulmonary TB among KVPs residing in hotspot settings in Ethiopia was 11.70%. This combined prevalence surpassed the findings of a previous large-scale study, which focused on investigating TB prevalence among KVPs in Ethiopia and reported a rate of 4.6% [3]. In addition, our finding was also higher than the first national survey report of Ethiopia [23]. This could be because our study included study reports from various high-risk settings, while the national report did not include congregate or high-risk settings in the poll. Dememew and his colleagues also excluded individuals residing in certain hotspot settings, such as spiritual holy water sites, and University students [3]. Our finding was also slightly higher than the pooled prevalence reports of TB among Ethiopian prison inmates, as evidenced by two meta-analyses conducted at different periods, which reported rates of 8.3% [10], and 9.8% [9]. This could be attributed to the fact that those two studies solely focused on one hotspot setting (prison), whereas our study incorporated reports from various hotspot settings conducted in various high-risk settings for TB transmission. However, our finding was lower than the reported prevalence of TB among prison inmates in Uganda (13.7%) [55], Paraguay (14.5%) [56], and the Democratic Republic of the Congo (17.7%) [57]. Geographical differences, overcrowding in various hotspot settings, diagnostic procedures, and inadequate ventilation in confined cells could all potentially explain the observed differences.

In this meta-analysis, we conducted a subgroup analysis to assess the magnitude of pulmonary TB across different hotspot settings. Thus, our study found the combined weighted prevalence of TB among prison inmates in Ethiopia to be 8.8%. Our findings were in line with the pooled prevalence results of two previous Ethiopian studies, reporting prevalences of 8.3% [10] and 9.8% [9], as well as the pooled prevalence report of TB in prison inmates in Sub-Saharan African countries (7.7%) [58]. However, it was lower than the reported rates in Uganda (13.7%) [55], Paraguay (14.5%) [56], and the Democratic Republic of the Congo (17.7%) [57]. This variation could be attributed to geographical disparities, differences in diagnostic methods, and variations in overcrowded living conditions among people residing in different hotspot settings, all of which could potentially explain the observed differences. Although there is limited information due to insufficient study reports to provide strong evidence, the subgroup analysis in this study revealed the pooled prevalence of pulmonary TB among University students (23.1%), refugees (28.4%), attendees of spiritual holy water sites (12.3%), healthcare workers (11.1%), and homeless individuals (5.8%). An earlier large-scale community-based study in Ethiopia, encompassing KVPs such as prisoners, internally displaced individuals, the

homeless, female sex workers, healthcare workers, internal migrant workers, and residents in missionary charities residing in hotspot settings, reported a TB prevalence of 4.6% [3]. Besides, an earlier systematic review study also found that the incidence and prevalence of TB among refugees and migrant populations ranged from 19 to 754 per 100,000 population and from 18.7 to 535 per 100,000 population, respectively [54]. Earlier studies indicated a significant burden of PTB among populations attending spiritual holy water sites in Ethiopia [16, 22], where these hotspot settings had not received much attention from the national TB prevention and control program. This underscores the significantly higher transmission of pulmonary TB and TB disease among population groups residing in hotspot settings or congregate settings, emphasizing the need for regional and national TB control programs to develop and implement targeted interventions.

Furthermore, the subgroup analysis in this meta-analysis, aimed at determining the overall magnitude of pulmonary TB before and post-2015 among the Ethiopian population residing in hotspot settings, revealed a declining trend. Before 2015, the prevalence of pulmonary TB was 14.04%, whereas it decreased to 10.79% post-2015. This decline in trend could be attributed to the global effort to end the TB epidemic and eradicate TB, Ethiopia has committed to significantly expanding its efforts, including the swift adoption of new tools and interventions for KVPs residing in hotspot settings, as well as implementing strategies to end TB. Another possible reason for the declining trends in PTB prevalence post-2015 could be the country's increased efforts in active TB screening after 2015. This might indicate a rise in the total number of screened populations, particularly among KVPs. From 2010 to 2020, the incidence of TB in Ethiopia decreased by an average of 5% per year [58].

The present study has some limitations. Non-English language reports were excluded from this systematic review and meta-analysis. The diverse TB diagnostic approaches utilized in the various studies also impacted the overall prevalence report. Furthermore, environmental conditions and transmission rates of pulmonary TB in different hotspot settings may vary, impacting the overall prevalence report. Besides, generalizing the results posed a challenge due to the limited information and data available from certain hotspot settings. Moreover, only studies conducted between 2000 and 2023 were considered for inclusion.

## Conclusions

The pooled prevalence of pulmonary TB among KVPs residing in hotspot settings or congregate settings in Ethiopia is notably higher than that reported in previous studies. Despite the limited data available to establish and generate strong evidence, the prevalence of TB among KVPs residing in various congregate settings in Ethiopia was higher. A declining trend in the prevalence of TB among KVPs in Ethiopia residing in high-risk settings was observed from 2000 to 20233. Thus, the national TB control programs should give due attention and appropriate control measures should be instituted that include regular systematic TB screening, compulsory TB testing for presumptive TB cases among KVPs, and tightened infection control at hotspot settings.

## Supporting information

**S1 File. Supporting tables and figures.**
(DOCX)

**S1 Checklist. PRISMA 2020 checklist.**
(DOCX)

## Author Contributions

**Conceptualization:** Melese Abate Reta, Ermias Getachew, Getinet Kumie, Marye Nigatie, Nontuthuko Excellent Maningi, P. Bernard Fourie.

**Data curation:** Melese Abate Reta, Zelalem Asmare, Assefa Sisay, Yalewayker Gashaw, Getinet Kumie, Marye Nigatie, Wagaw Abebe, Alene Geteneh, Atitegeb Abera Kidie, Biruk Beletew Abate, Nontuthuko Excellent Maningi, P. Bernard Fourie.

**Formal analysis:** Melese Abate Reta, Zelalem Asmare, Assefa Sisay, Yalewayker Gashaw, Ermias Getachew, Muluken Gashaw, Zelalem Dejazmach, Abdu Jemal, Solomon Gedfie, Getinet Kumie, Marye Nigatie, Wagaw Abebe, Agenagnew Ashagre, Tadesse Misganaw, Woldeteklehaymanot Kassahun, Selamyhun Tadesse, Alene Geteneh, Atitegeb Abera Kidie, Biruk Beletew Abate, Nontuthuko Excellent Maningi, P. Bernard Fourie.

**Funding acquisition:** Melese Abate Reta.

**Investigation:** Melese Abate Reta, Ermias Getachew.

**Methodology:** Melese Abate Reta, Zelalem Asmare, Assefa Sisay, Yalewayker Gashaw, Atitegeb Abera Kidie, Biruk Beletew Abate.

**Project administration:** Melese Abate Reta.

**Resources:** Melese Abate Reta.

**Software:** Melese Abate Reta, Zelalem Asmare, Assefa Sisay, Yalewayker Gashaw, Zelalem Dejazmach, Solomon Gedfie, Getinet Kumie, Marye Nigatie, Wagaw Abebe, Agenagnew Ashagre, Tadesse Misganaw, Woldeteklehaymanot Kassahun, Selamyhun Tadesse, Alene Geteneh, Atitegeb Abera Kidie, Biruk Beletew Abate, Nontuthuko Excellent Maningi, P. Bernard Fourie.

**Supervision:** Melese Abate Reta, Nontuthuko Excellent Maningi, P. Bernard Fourie.

**Validation:** Melese Abate Reta, Zelalem Asmare, Assefa Sisay, Yalewayker Gashaw, Abdu Jemal, Getinet Kumie, Marye Nigatie, Wagaw Abebe, Agenagnew Ashagre, Tadesse Misganaw, Woldeteklehaymanot Kassahun, Alene Geteneh, Atitegeb Abera Kidie, Biruk Beletew Abate, Nontuthuko Excellent Maningi, P. Bernard Fourie.

**Visualization:** Melese Abate Reta, Zelalem Asmare, Assefa Sisay, Yalewayker Gashaw, Abdu Jemal, Solomon Gedfie, Getinet Kumie, Marye Nigatie, Wagaw Abebe, Agenagnew Ashagre, Tadesse Misganaw, Woldeteklehaymanot Kassahun, Selamyhun Tadesse, Alene Geteneh, Atitegeb Abera Kidie, Biruk Beletew Abate, Nontuthuko Excellent Maningi, P. Bernard Fourie.

**Writing – original draft:** Melese Abate Reta, Zelalem Asmare, Assefa Sisay, Yalewayker Gashaw, Ermias Getachew, Muluken Gashaw, Zelalem Dejazmach, Abdu Jemal, Solomon Gedfie, Getinet Kumie, Marye Nigatie, Wagaw Abebe, Agenagnew Ashagre, Tadesse Misganaw, Woldeteklehaymanot Kassahun, Selamyhun Tadesse, Alene Geteneh, Atitegeb Abera Kidie, Biruk Beletew Abate, Nontuthuko Excellent Maningi, P. Bernard Fourie.

**Writing – review & editing:** Melese Abate Reta, Zelalem Asmare, Assefa Sisay, Yalewayker Gashaw, Ermias Getachew, Muluken Gashaw, Zelalem Dejazmach, Abdu Jemal, Solomon Gedfie, Getinet Kumie, Marye Nigatie, Wagaw Abebe, Agenagnew Ashagre, Tadesse Misganaw, Woldeteklehaymanot Kassahun, Selamyhun Tadesse, Alene Geteneh, Atitegeb Abera Kidie, Biruk Beletew Abate, Nontuthuko Excellent Maningi, P. Bernard Fourie.

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
