## [Decision Letter · Decision Letter 0]

16 Jul 2024

PONE-D-24-19020Prevalence of pulmonary tuberculosis among key populations in hotspot settings in Ethiopia: A systematic review and Meta-AnalysisPLOS ONE

Dear Dr. Reta,

Thank you for submitting your manuscript to PLOS ONE. After careful consideration, we feel that it has merit but does not fully meet PLOS ONE’s publication criteria as it currently stands. Therefore, we invite you to submit a revised version of the manuscript that addresses the points raised during the review process.

Congratulations again for coming up with this quite relevant evidence of PTB among KVP. This is very important for Ethiopian TB program where the focus is on tailored active TB case finding is among KVPs at hotspot settings for TB.

However, the is one critical comment to deal with. It is very instrumental to revise that all the literatures/references underwent for review were PTB. That is, it is advisable to check your topic/objective of assessing the prevalence PTB versus the cited references—looks some publications are not categorized into form/types of TB ,i.e  not limited to PTB by excluding EPTB. If you are interested in only PTB that may be reflected clearly in the literature search and inclusion/exclusion criteria.

Also, here are other minor comments for your consideration.

The document will benefit out of professional English language editorial (grammar, structure and spacing) E.g university student vs University student; spacing in “..8.8% (95%CI: 5.00-12.55), University students 23.1% (95%CI: 15.81-30.37), Refugee camps 28.4%(95%CI: -1.27-58.15), Homeless individuals 5.8% (95%CI: -0.67- 12.35),..”Check the recently revised terminology ( under STOP TB KVP) as  ‘key and vulnerable population’(KVPs) is the updated and standard  terminology in use.You may use and refer the latest TB incidence estimate of TB in Ethiopia, 126/100k--Global tuberculosis report 2023 (who.int) in the statement “ Ethiopia’s TB incidence was 157 per 100,000 population, in 2020 [5].” Consider changing the referenced in text and list as well.Check the explanation and justification of the declining in the prevalence of KVP. This might show the increasing effort of active TB screening after 2015 where the denominator, # screened KVPs, might be increasing.Review the manuscript write up per the PLOS ONE writing standard, e.g. TB   cannot be defined twice- at introduction and discussion.

We look forward to receiving your revised manuscript.

Kind regards,

Zewdu Gashu Dememew, M.D, PhD

Academic Editor

PLOS ONE

Additional Editor Comments (if provided):

Reviewers' comments:

Reviewer's Responses to Questions

**Comments to the Author**

1. Is the manuscript technically sound, and do the data support the conclusions?

Reviewer #1: Yes

Reviewer #2: Yes

2. Has the statistical analysis been performed appropriately and rigorously? 

Reviewer #1: Yes

Reviewer #2: Yes

3. Have the authors made all data underlying the findings in their manuscript fully available?

Reviewer #1: Yes

Reviewer #2: Yes

4. Is the manuscript presented in an intelligible fashion and written in standard English?

Reviewer #1: Yes

Reviewer #2: Yes

5. Review Comments to the Author

Reviewer #1: The manuscript is scientifically sound. The aim of the study corresponds to the methods used. The protocol is registered in International Prospective Register of Systematic Reviews (PROSPERO) (ID: 543925). For the literature review the PRIZM system was used. The statistical analysis is transparent and clear. The article is well structured with respect of all parts of the research.

Reviewer #2: The manuscript provided important information of TB prevalence in risk groups in Ethiopia. It is very interesting that university students were considered a risk group along side refugees and prison inmates and what is even more surprising that TB prevalence was the highest in that group compared to all others. It would be very interesting to hear authors ideas on that issue in discussion - why students are considered a risk group and why there is so much TB. Additionaly authors could show some firgures on risk groups' contribution to overall burden of TB in Ethiopia percentage wise, if this data is available, e.g. total number of cases per year in risk groups and outside of risk groups.

Also minor errors occur thoughout the text, e.g. s ".....observed from 2000 to 20233. Thus.....' etc

6. PLOS authors have the option to publish the peer review history of their article (what does this mean?). If published, this will include your full peer review and any attached files.

Reviewer #1: No

Reviewer #2: No

---

## [Author Response · Author response to Decision Letter 0]

7 Aug 2024

Author's Response to Reviewers’ comments

Title: Prevalence of pulmonary tuberculosis among key populations in hotspot settings of Ethiopia. A Systematic Review and Meta-Analysis

Manuscript ID: PONE-D-24-19020

PLOS ONE

Dear reviewers

First and foremost, we would like to extend our deepest gratitude to the reviewer(s)/editors for their time, effort, and commitment to reviewing our submitted paper and for their constructive comments. We greatly appreciate the detailed and thoughtful comments provided to our manuscript. These insightful comments were instrumental in guiding us to refine and improve our manuscript. They provided valuable perspectives that helped us address key issues and strengthen the overall presentation of our research. The detailed feedback highlighted areas that needed further clarification and allowed us to make necessary revisions, ultimately resulting in a more robust and impactful manuscript. We have made a rigorous effort to amend our manuscript in accordance with the reviewers'/editor’s comments and have employed additional revisions to further enhance the quality of our work. As highlighted in the main document, we made significant revisions in the manuscript. Hence, we made sure that each one of the comments has been addressed carefully and the manuscript is revised accordingly. Below are detailed responses (point-by-point responses) to all the comments/suggestions. All revisions that have been made to our manuscript are highlighted (track changing) in the revised versions of the manuscript. Please let us know if you have further inquiries regarding the revised manuscript or if any queries or comments still need clarity. We will be happy to address them, now on time.

Thank you very much for your time and efforts to evaluate our manuscript.

With sincere regards, 

Melese Abate Reta (PI)

Date: 03 August 2024

Reviewer’s Comments and Author/s’ Responses

Major comment

1. Reviewer’s/editor’s Comment_1: 

However, there is one critical comment to deal with. It is very instrumental to revise that all the literatures/references underwent for review were PTB. That is, it is advisable to check your topic/objective of assessing the prevalence PTB vs the cited references—looks some publication are not categorized in to form/types of TB, i.e not limited to PTB by excluding EPTB. If you are interested in only PTB that may be reflected clearly in the literature search and inclusion/exclusion criteria.

Author’s response_1: 

We greatly appreciate your valuable comments on this point. As we have clearly highlighted in the “title” of our paper, “inclusion criteria”, “search strategies”, and the “study outcomes” section, our main focus/objective is to assess the combined/pooled prevalence of PTB while excluding extrapulmonary TB (EPTB). We carefully constructed our search terms for PTB to ensure we captured all relevant papers from the electronic databases. However, if the eligible and included studies report both the incidence or prevalence of PTB and EPTB among presumptive TB patients, we carefully extract the data pertaining to PTB cases only and calculate it’s prevalence by excluding EPTB cases. We emphasize this statement in the “Study Outcomes” section of the manuscript. Additionally, in our exclusion criteria: (c) "Studies reporting only extrapulmonary TB cases or latent TB" were excluded. 

Minor Comments

2. Reviewer’s Comment_2: 

The document will benefit out of professional English language editorial (grammar, structure and spacing) E.g university student vs University student; spacing in “..8.8% (95%CI: 5.00-12.55), University students 23.1% (95%CI: 15.81-30.37), Refugee camps 28.4%(95%CI: -1.27-58.15), Homeless individuals 5.8% (95%CI: -0.67- 12.35),..”

Author’s response_2: 

Thank you very much for your constructive comments. We have revised the paper to improve its writing, including grammar, structure, and spacing, throughout the manuscript. Additionally, we invited native English speakers to review our paper, resulting in many corrections.

3. Reviewer’s Comment_3: 

Check the recently revised terminology (under STOP TB KVP) as ‘key and vulnerable population’(KVPs) is the updated and standard terminology in use.

 Author’s response_3: 

Thank you very much. We have reviewed the recently revised terminology for KVP and updated it to "key and vulnerable populations" (KVPs). These corrections have been made throughout the document.

4. Reviewer’s Comment_4: 

You may use and refer the latest TB incidence estimate of TB in Ethiopia, 126/100k--Global tuberculosis report 2023 (who.int) in the statement “ Ethiopia’s TB incidence was 157 per 100,000 population, in 2020 [5].” Consider changing the referenced in text and list as well.

 Author’s response_4: 

Thank you for your valuable comment. We have referred to the latest TB incidence estimates for Ethiopia from the Global Tuberculosis Report 2023 (126/100,000) and updated/changed the reference accordingly.

5. Reviewer’s Comment_5: 

Check the explanation and justification of the declining in the prevalence of KVP. This might show the increasing effort of active TB screening after 2015 where the denominator, # screened KVPs, might be increasing.

 Author’s response_5: 

We have reviewed the explanation and justification for the declining trends in the prevalence of KVP and included the following statement in the discussion section: “Another possible reason for the decline in PTB prevalence post-2015 could be the country’s increased efforts in active TB screening after 2015. This might reflect a rise in the total number of screened populations, especially among key and vulnerable groups.”

6. Reviewer’s Comment_6: 

Review the manuscript write up per the PLOS ONE writing standard, e.g. TB cannot be defined twice- at introduction and discussion.

Author’s response_6: 

Thank you for this important comment. We have tried to adhere to the PLOS ONE manuscript writing standards and revised the introduction and discussion sections to remove the repeated use of the abbreviation "TB".

---

## [Editor Report · Decision Letter 1]

13 Aug 2024

Prevalence of pulmonary tuberculosis among key and vulnerable populations in hotspot settings of Ethiopia. A Systematic Review and Meta-Analysis

PONE-D-24-19020R1

Dear Dr. Melese,

We’re pleased to inform you that your manuscript has been judged scientifically suitable for publication and will be formally accepted for publication once it meets all outstanding technical requirements.

Kind regards,

Zewdu Gashu Dememew, M.D

Academic Editor

PLOS ONE

Dear Dr Melse,

Congratulations again for this important evidence, and for revising your article per the given comment.

I am happy that you also consider a few and very minor suggestion before publication.

• ‘’compulsory TB testing’’ (for whom? Presumptive TB cases among KVPs or all KVPs irrespective of TB signs and symptoms?): check if this is a feasible and practical recommendation in Ethiopia.

• Figure 1: the Asterix ( *) refers to what?

• Once defined during the first appearance, ‘Key and Vulnerable Population (KVP)’ could be written as KVP throughout.

Best regards

---

## [Editor Report · Acceptance letter]

20 Aug 2024

PONE-D-24-19020R1 

PLOS ONE

Dear Dr. Reta, 

I'm pleased to inform you that your manuscript has been deemed suitable for publication in PLOS ONE. Congratulations! Your manuscript is now being handed over to our production team.

Kind regards, 

on behalf of

Dr. Zewdu Gashu Dememew 

Academic Editor

PLOS ONE